# AICDA drives epigenetic heterogeneity and accelerates germinal center-derived lymphomagenesis

Matt Teater[1,2], Pilar M. Dominguez[1], David Redmond[2], Zhengming Chen[3], Daisuke Ennishi[4], David W. Scott[4], Luisa Cimmino[5], Paola Ghione[1,6], Jayanta Chaudhuri[7], Randy D. Gascoyne[8], Iannis Aifantis[5], Giorgio Inghirami[9], Olivier Elemento[2,10], Ari Melnick[1] & Rita Shaknovich[1,11]

Epigenetic heterogeneity is emerging as a feature of tumors. In diffuse large B-cell lymphoma (DLBCL), increased cytosine methylation heterogeneity is associated with poor clinical outcome, yet the underlying mechanisms remain unclear. Activation-induced cytidine deaminase (AICDA), an enzyme that mediates affinity maturation and facilitates DNA demethylation in germinal center (GC) B cells, is required for DLBCL pathogenesis and linked to inferior outcome. Here we show that AICDA overexpression causes more aggressive disease in BCL2-driven murine lymphomas. This phenotype is associated with increased cytosine methylation heterogeneity, but not with increased AICDA-mediated somatic mutation burden. Reciprocally, the cytosine methylation heterogeneity characteristic of normal GC B cells is lost upon AICDA depletion. These observations are relevant to human patients, since DLBCLs with high AICDA expression manifest increased methylation heterogeneity vs. AICDA-low DLBCLs. Our results identify AICDA as a driver of epigenetic heterogeneity in B-cell lymphomas with potential significance for other tumors with aberrant expression of cytidine deaminases.

[1] Department of Medicine, Division of Hematology and Medical Oncology, Weill Cornell Medicine, New York, NY 10021, USA. [2] Institute for Computational Biomedicine, Department of Physiology and Biophysics, Weill Cornell Medicine, New York, NY 10021, USA. [3] Division of Biostatistics and Epidemiology, Weill Cornell Medicine, New York, NY 10021, USA. [4] Centre for Lymphoid Cancer, British Columbia Cancer Agency, Vancouver, BC V5Z 4E6, Canada. [5] Department of Pathology, Laura and Isaac Perlmutter Cancer Center, and The Helen L. and Martin S. Kimmel Center for Stem Cell Biology, NYU School of Medicine, New York, NY 10016, USA. [6] Division of Hematology, Department of Experimental Medicine and Oncology, University of Turin, 10124 Turin, Italy. [7] Immunology Program, Memorial Sloan-Kettering Cancer Center, Gerstner Sloan-Kettering Graduate School, New York, NY 10021, USA. [8] Department of Pathology, British Columbia Cancer Agency, Vancouver, BC V5Z 4E6, Canada. [9] Pathology and Laboratory Medicine Department, Weill Cornell Medicine, New York, NY 10021, USA. [10] Caryl and Israel Englander Institute for Precision Medicine, Weill Cornell Medicine, New York, NY 10021, USA. [11] Cancer Genetics, Inc., Rutherford, NJ 07070, USA. Matt Teater and Pilar M. Dominguez contributed equally to this work. Correspondence and requests for materials should be addressed to O.E. (email: ole2001@med.cornell.edu) or to A.M. (email: amm2014@med.cornell.edu) or to R.S. (email: rshaknovich@gmail.com)

Epigenetic mechanisms, including aberrant cytosine methylation patterning, play critical roles in the pathogenesis and progression of lymphoid malignancies and other tumors[1]. As opposed to genomic DNA sequences, cytosine methylation distribution has great plasticity and can be dynamically redistributed in response to environmental changes or due to the influence of transcription regulatory factors[2]. One consequence of plasticity is the variable capacity of tumors to diverge from their cell of origin, which manifests as inter-tumor and intra-tumor epigenetic heterogeneity. Inter-tumor and intra-tumor heterogeneity have been documented to occur in lymphoid and myeloid malignancies and are associated with inferior clinical outcome[3–7]. Intra-tumor heterogeneity is proposed to increase fitness in tumors by providing individual cells with options for adaptation to changing environmental conditions or exposure to therapeutic agents[8, 9]. Given the clinical and biological significance of aberrant epigenetic plasticity in tumors, it is of critical importance to identify mechanisms that mediate this phenomenon.

One possible source of plasticity lies in proteins that modify the epigenome. In the context of lymphomas, one such putative factor is the cytosine deaminase activation-induced cytidine deaminase (AICDA), which is specifically expressed and active in germinal center-like B cells (GCB), the cells of origin of diffuse large B-cell lymphoma (DLBCL). In the normal GC reaction, AICDA is required to induce somatic hypermutation (SHM) and class-switch recombination of the immunoglobulin (Ig) loci. Off-target AICDA-induced mutations, affecting non-Ig genes (e.g., *BCL6* locus), may contribute to lymphomagenesis[10]. However, in addition to somatic mutation, AICDA also can alter the epigenome by deaminating methyl-cytosines, which are then putatively repaired with unmethylated nucleotides[11, 12]. Indeed, GC B cells normally exhibit extensive loss of cytosine methylation along with more heterogeneous methylation patterning, effects shown to be largely dependent on AICDA[13]. Here, we demonstrate that AICDA is a key driving force in generating cytosine methylation heterogeneity in GC B cells and GC-derived lymphomas, an effect that could lead to increased tumor fitness and more aggressive disease. We also find that AICDA expression may be among the contributing factors to both the inferior outcome and greater epigenetic heterogeneity observed in the activated B-cell-like (ABC) subtype of DLBCL.

## Results

**AICDA overexpression accelerates BCL2-driven lymphoma.** To characterize the actions of AICDA in GC-derived lymphomagenesis, we transplanted bone marrow cells from VavP-*Bcl2* transgenic mice transfected with AICDA-expressing retrovirus (VavP-*Bcl2+Aicda*) or empty vector control (VavP-*Bcl2*) into lethally irradiated recipients. All mice were sacrificed after 8 months, the timepoint at which VavP-*Bcl2+Aicda* animals presented signs of morbidity. Histopathological examination revealed more aggressive disease in VavP-*Bcl2+Aicda* ($n = 8$) than in VavP-*Bcl2* ($n = 7$) mice, with greater disruption of splenic architecture and neoplastic B-cell expansion in organs such as the lung, liver, and kidney (Fig. 1a and Supplementary Fig. 1a). Reminiscent of human DLBCLs, the spleens of VavP-*Bcl2+Aicda* but not VavP-*Bcl2* control mice exhibited white pulp expansion with replacement by sheets of neoplastic B cells (Fig. 1a). Polymerase chain reaction (PCR) analysis of the Ig heavy-chain variable region (IgVH) of B220+ splenocytes identified clonal rearrangements, consistent with lymphoma (Supplementary Fig. 1b). The VavP-*Bcl2+Aicda* neoplastic cells were larger, exhibited greater pleomorphic morphology, and had higher Ki67 positivity (Fig. 1b), indicating increased number of proliferating cells, a feature that is correlated with more

aggressive DLBCL[14]. Necropsy of diseased animals revealed greater burden of disease in the spleen, lung, kidney, and livers of VavP-*Bcl2+Aicda* animals, as quantified by the degree of neoplastic expansion and infiltration (Fig. 1c). Western blotting of isolated splenic cells confirmed the presence of AICDA protein in all the samples, indicating GC origin of the tumors, with generally higher levels of AICDA observed in VavP-*Bcl2+Aicda* mice than VavP-*Bcl2* control mice (Supplementary Fig. 1c, e). Immunophenotyping of diseased spleens showed a high proportion of B220+CD95+GL7int/high cells in all the mice, confirming that lymphomas were GC derived (Supplementary Fig. 1d). As expected, the abundance of GC-derived tumor cells was significantly higher in the VavP-*Bcl2+Aicda* given their increased tumor burden. In addition, we observed a higher proportion of B cells expressing IgG1 in VavP-*Bcl2+Aicda* mice compared to VavP-*Bcl2* mice, suggesting increased class switching in the presence of excess AICDA (Supplementary Fig. 1d). A separate cohort of mice was followed longitudinally to assess impact of AICDA expression on survival. VavP-*Bcl2+Aicda* mice ($n = 10$) manifested significantly shortened lifespan (log-rank test $P = 0.0289$), with a median survival of 214 days as opposed to 293 days for VavP-*Bcl2* ($n = 9$) animals (Fig. 1d). Overall, our data indicate that AICDA overexpression is associated with a more aggressive disease phenotype and decreased survival.

**AICDA induces DNA methylation heterogeneity in lymphomas.** To explore whether *Aicda*-overexpressing lymphomas manifested differences in cytosine methylation distribution, we performed enhanced reduced representation bisulfite sequencing (ERRBS), a quantitative single-nucleotide resolution methylome sequencing technique[15], on VavP-*Bcl2+Aicda* ($n = 7$) and VavP-*Bcl2* ($n = 6$) lymphoma B-cell populations. Initial examination of global methylation levels indicated that VavP-*Bcl2+Aicda* tumors showed fewer highly methylated cytosine-guanine dinucleotides (CpGs), consistent with the role of AICDA in demethylation (Fisher's exact test $P = 2.27e{-}14$; Supplementary Fig. 2a). To interrogate whether VavP-*Bcl2+Aicda* tumors also exhibited more diverse cytosine methylation patterning, we examined pairwise methylation distance between cytosine methylation profiles and found that VavP-*Bcl2+Aicda* lymphomas indeed manifested greater global inter-tumor heterogeneity than their VavP-*Bcl2* counterparts (Fig. 2a; Wilcoxon's $P = 0.00112$). Since AICDA induces DNA hypomethylation in GC B cells[13], we postulated that the AICDA-mediated DNA methylation heterogeneity would be more prominently observed among hypomethylated cytosines in VavP-*Bcl2+Aicda* lymphoma cells. We therefore evaluated AICDA-associated changes in two dimensions: change in mean methylation level and the change in "spread" of methylation values, measured as interquartile range (IQR) between respective tumors, at each CpG showing >20% methylation difference. Among these selected CpGs, inter-tumor heterogeneity was significantly associated with hypomethylation in VavP-*Bcl2+Aicda* mice compared to control mice (Fisher's exact test $P = 3.68e{-}40$; Fig. 2b).

To identify AICDA DNA methylation heterogeneity signatures in a more comprehensive and unbiased manner, we performed a principal component analysis (PCA) on all CpGs according to the two dimensions of AICDA-associated changes: mean methylation and IQR (Supplementary Fig. 2b). This analysis showed that, consistent with the results from Fig. 2b, the most pronounced alteration to the entire VavP-*Bcl2+Aicda* methylome involved methylation loss and increased inter-tumor heterogeneity (Supplementary Fig. 2c). From this approach, we identified a VavP-*Bcl2+Aicda* cytosine methylation signature containing 49,750

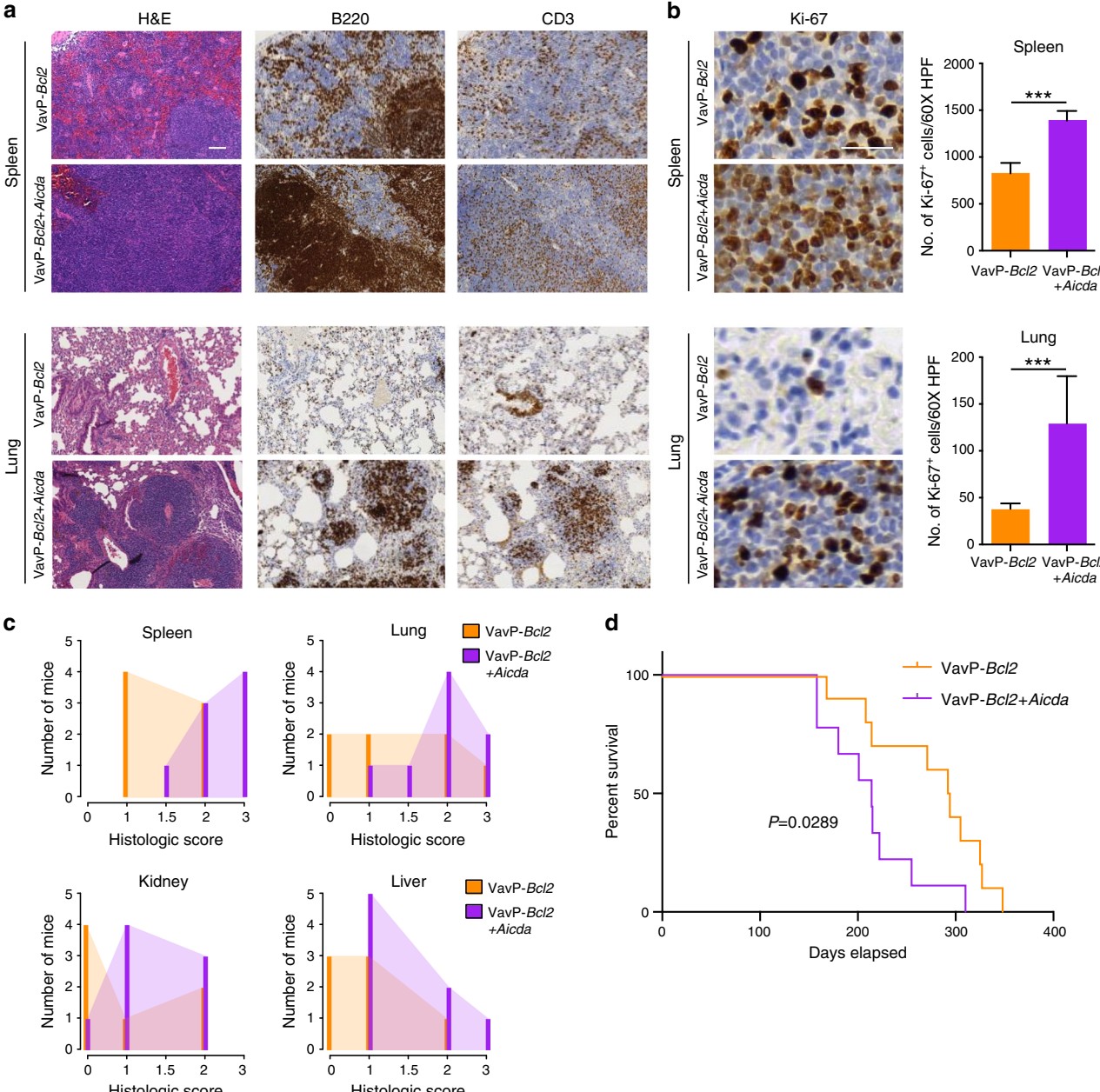

**Fig. 1** AICDA overexpression results in more aggressive BCL2-driven lymphomas. **a** Representative histologic sections of formalin-fixed, paraffin-embedded spleens and lung tissues from VavP-*Bcl2* and VavP-*Bcl2+Aicda* mice. Sections were stained with H&E and antibodies specific for B220 and CD3. Scale bar represents 100 µm **b** Representative histologic sections stained with anti-Ki67 (left) and quantification of Ki67+ cells (right) in the spleen and lung tissues from VavP-*Bcl2* and VavP-*Bcl2+Aicda* mice. Bars represent mean number of Ki67+ cells in 10 fields of spleen and lung sections and error bars indicate standard deviation; scale bar represents 100 µm; two-tailed *t* test ***$P < 0.001$. **c** Histologic score, measuring relative organ infiltration, of spleen, lung, kidney, and liver from VavP-*Bcl2* ($n = 7$) and VavP-*Bcl2+Aicda* ($n = 8$) mice. Scores correspond to no (0), mild (1), moderate (2), and marked (3) infiltration by neoplastic lymphocytes. **d** Kaplan–Meier survival curve of VavP-*Bcl2* ($n = 10$) and VavP-*Bcl2+Aicda* ($n = 9$) mice. Significant differences in survival were evaluated by log-rank (Mantel–Cox) test

CpGs (VavP-*Bcl2* AICDA-perturbed CpGs; Supplementary Fig. 2c). This signature was comprised of CpGs that lost methylation, primarily from a highly methylated state within VavP-*Bcl2* controls (Fig. 2c). The signature CpGs also exhibited increased inter-tumor heterogeneity among the VavP-*Bcl2+Aicda* lymphomas, compared to relatively stable methylation states within VavP-*Bcl2* controls (median VavP-*Bcl2+Aicda* IQR = 19.2%; median VavP-*Bcl2* IQR = 5.56%; Fig. 2d). Notably, the loss of methylation of this VavP-*Bcl2+Aicda* cytosine methylation signature results in an increase in intra-tumor heterogeneity

compared to the VavP-*Bcl2* tumors (paired Wilcoxon's test $P < 1e-300$; Fig. 2e). VavP-*Bcl2+Aicda*-perturbed CpGs were significantly depleted from gene promoters, but were highly enriched within introns and intergenic regions (Fig. 2f; Fisher's exact test $P < 0.001$). VavP-*Bcl2+Aicda*-perturbed CpGs were also largely depleted within CpG islands, but were slightly enriched within CpG "shores" and "shelves" (Fig. 2g; Fisher's exact test $P < 0.001$). These AICDA-mediated epigenetic changes are unlikely to stem from differences in the proliferation rate between AICDA-overexpressing and control lymphomas since DNA methylation

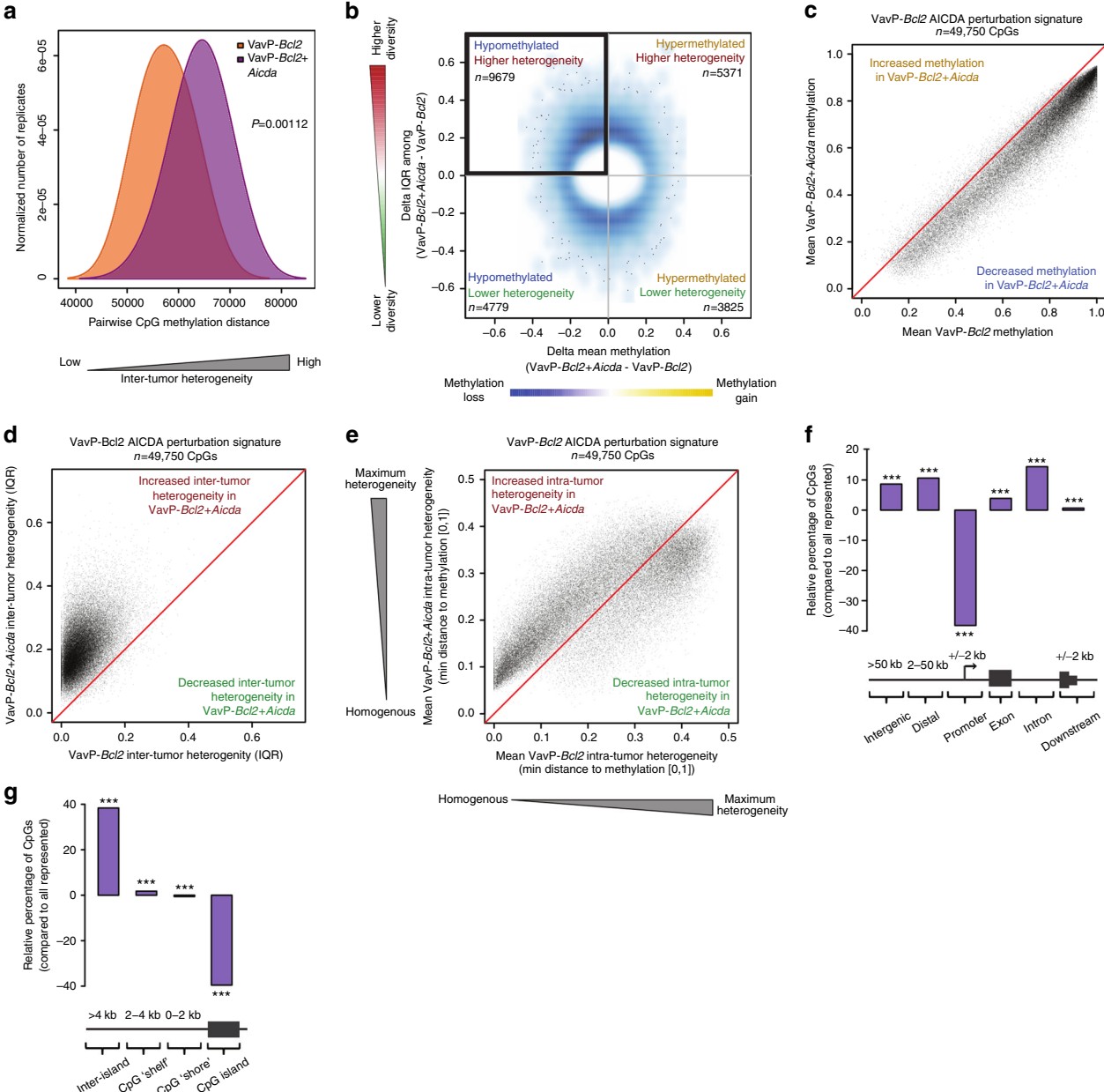

**Fig. 2** AICDA overexpression induces DNA methylation heterogeneity and hypomethylation in VavP-*Bcl2+Aicda* tumors. **a** Density plot showing the inter-tumor pairwise methylation distance between ERRBS profiles of VavP-*Bcl2* and VavP-*Bcl2+Aicda* tumors. VavP-*Bcl2+Aicda* tumors have greater pairwise distance, indicating increased inter-tumor heterogeneity among methylation profiles; two-sided Wilcoxon's signed-rank test. **b** Density scatterplot showing the relationship between methylation change (*x*-axis) and change in inter-tumor diversity (*y*-axis) for all CpGs manifesting >20% combined methylation level difference and/or IQR difference. **c–e** Scatterplots showing shift in mean methylation values (**c**), inter-tumor diversity (**d**), and intra-tumor heterogeneity (**e**) of AICDA perturbation signature between VavP-*Bcl2* and VavP-*Bcl2+Aicda*. **f** Bar plot showing the distribution of AICDA-perturbed CpGs relative to the distribution of all represented CpGs; Fisher's exact test. **g** Bar plot showing the relative distribution of AICDA-perturbed CpGs within proximity to CpG islands; Fisher's exact test (\*$P < 0.05$, \*\*$P < 0.01$, \*\*\*$P < 0.001$)

heterogeneity in lymphoma cells is independent of the mitotic ratio, as we previously showed[3]. Additionally, as the retroviral vector transfection of bone marrow will express of AICDA in all hematopoietic compartments, we analyzed ERRBS profiles of other differentiated cell lineages including myeloid Gr1[+] cells and T cells isolated from the spleens of transplanted mice. There was no evidence of extensive hypomethylation or increased epigenetic heterogeneity in these non-B-cell lineages (Supplementary Figure 3a–c). This suggests that GC derived B cells are more tolerant of the actions of AICDA than other lineages and hence the observed epigenetic effects may be specific to GC-derived cells

and are not inherited from earlier stages of hematopoietic development.

AICDA overexpression could also contribute to lymphoma-genesis via introduction of somatic mutations at non-Ig loci, such as proto-oncogenes. To evaluate the contribution of AICDA mutational activity to the aggressive phenotype observed in VavP-*Bcl2+Aicda* mice, we first assessed the mutational burden of known Ig ($J_H4$ and $S\mu$) and non-Ig loci (*Bcl6*, *Cd83*, and *Pax5*) using targeted resequencing. We observed no significant differences in burden of nonsynonymous mutations (Supplementary Fig. 4a) or indels (Supplementary Fig. 4b) within Ig loci,

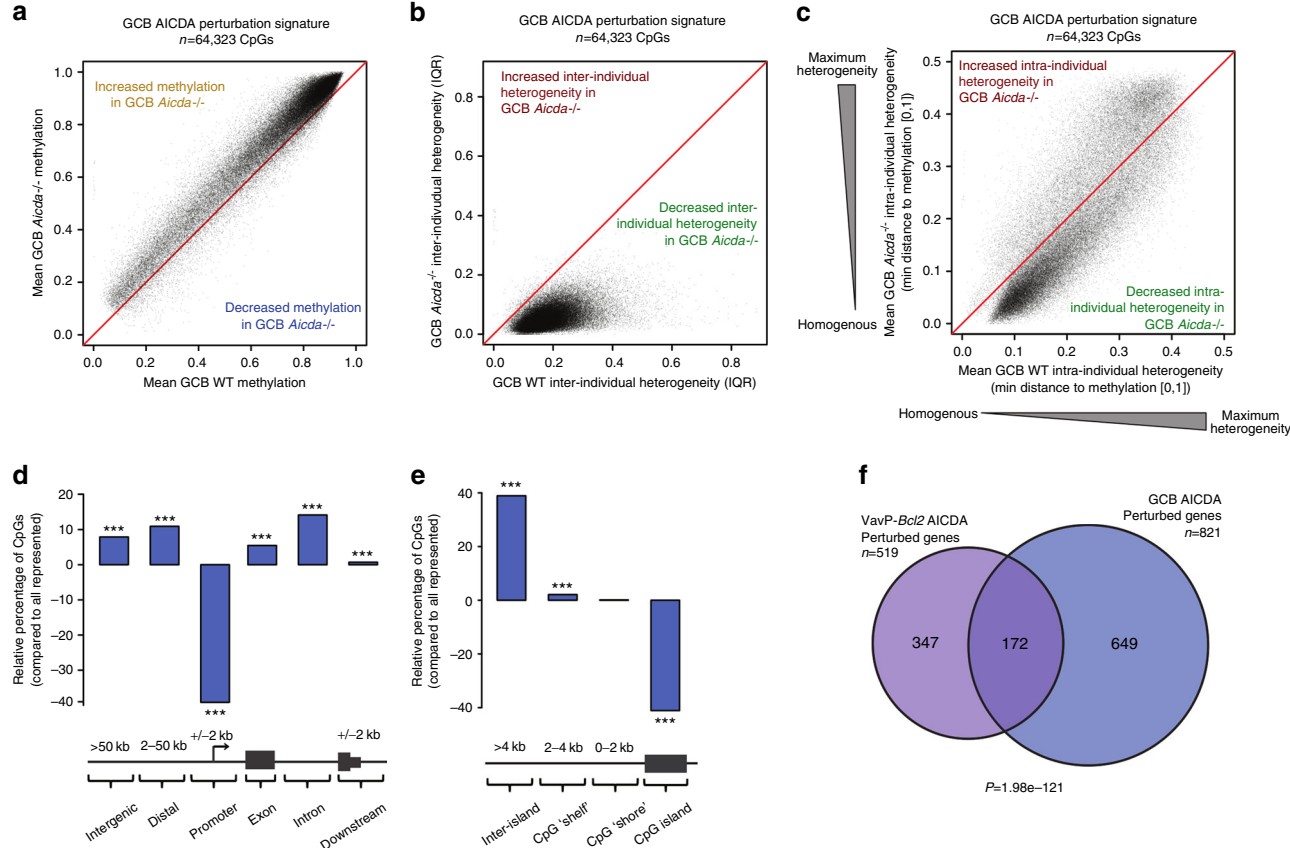

**Fig. 3** Loss of AICDA in GC B cells reduces DNA methylation heterogeneity and causes relative gain in methylation. **a–c** Scatterplots showing shift in mean methylation values (**a**), inter-individual methylation heterogeneity (**b**), and intra-individual methylation heterogeneity (**c**) of GC $Aicda^{-/-}$ perturbation signature between $Aicda^{-/-}$ and wild-type GC B cells. **d** Bar plot showing the distribution of GC $Aicda^{-/-}$ perturbed CpGs relative to the distribution of all represented CpGs; Fisher's exact test. **e** Bar plot showing the relative distribution GC $Aicda^{-/-}$ perturbed CpGs within proximity to CpG islands; Fisher's exact test. **f** Venn diagram showing the overlap between genes significantly over-representing GC $Aicda^{-/-}$ signature CpGs and genes over-representing VavP-*Bcl2*+*Aicda* signature CpGs; hypergeometric test. (*$P < 0.05$, **$P < 0.01$, ***$P < 0.001$)

*Cd83*, or *Pax5*, although indels were enriched at the *Bcl6* locus in VavP-*Bcl2*+*Aicda* mice (Supplementary Fig. 4b). To further explore genomic effects of AICDA overexpression, we performed whole exome sequencing in VavP-*Bcl2*+*Aicda* ($n = 4$) and VavP-*Bcl2* lymphomas ($n = 4$) and found no significant difference in nonsynonymous or indel mutation burdens (Supplementary Fig. 4c, d). These data suggest that the more aggressive VavP-*Bcl2*+*Aicda* phenotype is not significantly associated with increased mutational burden. Hence, overexpression of AICDA in B-cell lymphomas induces increased burden of epigenetic changes, without significant change in the burden of somatic mutation compared to lymphomas with endogenous AICDA levels, recapitulating earlier evidence that the level of AICDA expression did not increase aberrant SHM in DLBCL[16].

**AICDA loss in GCs attenuates DNA methylation heterogeneity.** DLBCLs arise from GC B cells, which normally express AICDA. As compared to their naïve, mature B-cell precursors, GC B cells manifest loss of cytosine methylation (i.e., relative hypomethylation)[17–19]. We analyzed the cytosine methylation profiles of $Aicda^{-/-}$ and wild-type (WT) GC B cells to explore a potential role of AICDA in epigenetic plasticity. Using PCA to assess inter-individual heterogeneity, we identified a GC $Aicda^{-/-}$ methylation signature consisting of 64,323 CpGs (GC B-cell $Aicda^{-/-}$-perturbed CpGs; Supplementary Fig. 5a, b). This CpG signature showed a inverse and reciprocal pattern to AICDA

overexpression: the CpGs exhibited relative gain of methylation (Fig. 3a), decreased inter-individual DNA methylation heterogeneity (Fig. 3b), and a relative loss of intra-individual methylation heterogeneity (paired Wilcoxon's $P < 1e-300$; Fig. 3c) in $Aicda^{-/-}$ vs. WT mice. Similar to the AICDA-perturbed CpGs in VavP-*Bcl2*+*Aicda*, these $Aicda^{-/-}$-perturbed CpGs were most enriched within introns and intergenic regions and depleted from promoters (Fig. 3d; Fisher's exact test $P < 0.001$). Additionally, GC $Aicda^{-/-}$ signature CpGs were under-represented in CpG islands and slightly enriched within CpG "shelves" (Fig. 3e; Fisher's exact test $P < 0.001$). Analysis of cell cycle in GC B cells showed no differences between WT and $Aicda^{-/-}$ cells (Supplementary Fig. 5c), suggesting that the AICDA-mediated epigenetic signature observed in GC B cells was not due to different rate of proliferation between WT and AICDA-deficient cells. Given the reciprocal nature of epigenetic effects between *Aicda* loss and overexpression, respectively, in normal and malignant B cells, we investigated whether AICDA preferentially affected the same genes within the two systems. Indeed, there was statistically significant overlap between genes enriched for VavP-*Bcl2* tumor AICDA signature CpGs within gene bodies and genes over-representing GC $Aicda^{-/-}$ signature CpGs within gene bodies (hypergeometric test $P = 1.98e-121$, Fig. 3f). Pathway analysis of the common AICDA-perturbed genes revealed consistent enrichment for gene signatures relevant to GC maintenance, including GC B-cell bivalent genes and signatures associated with GC exit. We also found enrichment for pathways relevant to GCs

and lymphoma, including signatures associated with Myc, p53, Bcl6, hypoxia, and GC exit (Supplementary Fig. 5d). In addition, we found AICDA-perturbed genes to include several recurrent aberrant SHM targets, such as DTX1, IRF8, EBF1, SYK, and PAX5. The magnitude of epigenetic heterogeneity was greater in the malignant cells (median AICDA signature VavP-*Bcl2*+*Aicda* IQR: 19.2%; median AICDA signature GCB WT IQR: 17.35%; Wilcoxon's $P < 1e−300$), which is expected since AICDA expression is transient in normal GCs but presumably sustained for longer periods in the lymphomas.

**High AICDA associates with epigenetic heterogeneity in DLBCL.** To investigate whether AICDA cytosine methylation signatures can be detected in primary human lymphomas, we performed ERRBS and RNA sequencing (RNAseq) in a cohort of 63 primary DLBCL patients. We compared cases with the highest AICDA expression (>50 counts per million reads (CPM); AICDA-high DLBCL; $n = 10$) against the cases with lowest or no AICDA expression (<2 CPM; AICDA-low DLBCL; $n = 19$, Supplementary Fig. 6a). Similar to mouse VavP-*Bcl2*+*Aicda* lymphomas, we found that AICDA-high DLBCL patients manifested greater global inter-tumor heterogeneity than AICDA-low cases (Fig. 4a; Wilcoxon's $P = 4.17e−06$). Given the concordance of these results with our previous murine findings, we performed a PCA to determine if higher levels of AICDA would have similar consequences upon the DLBCL methylome. Indeed, AICDA-high DLBCLs exhibited a pattern of increased inter-tumor heterogeneity and hypomethylation compared to DLBCL profiles with low expression of AICDA (Supplementary Fig. 6b). From this PCA, we identified a DLBCL AICDA-high methylation signature ($n = 37,557$ DLBCL AICDA-perturbed CpG; Supplementary Fig. 6c) that exhibited similar methylation dynamics to VavP-*Bcl2* +*Aicda* (and opposite to *Aicda*[−/−] GC B cells), including loss of CpG methylation from intermediate and highly methylated states (Fig. 4b) alongside gain of inter-tumor methylation heterogeneity (Fig. 4c). We also found that the AICDA-high DLBCLs had higher intra-tumor methylation heterogeneity compared to their lower AICDA-expressing controls (paired Wilcoxon's $P = 5.22e−229$; Fig. 4d). We found that CpGs perturbed in AICDA-high DLBCL profiles were depleted in gene promoters and enriched in exons, introns, and distal regions (Fig. 4e; Fisher's exact test $P < 0.001$). AICDA-high DLBCL perturbed CpGs were also under-represented in CpG islands and enriched in CpG "shelves" and "shores" (Fig. 4f; Fisher's exact test $P < 0.001$). Notably, we found significant overlap of gene orthologs affected by AICDA-perturbed CpGs in human AICDA-high DLBCLs and murine VavP-*Bcl2*+AICDA lymphomas (Fig. 4g; hypergeometric $P = 2.21e−23$) and with the gene orthologs affected by AICDA in normal GC B cells (Supplementary Fig. 6d; $P = 8.48e−33$). The identity of genes over-representing AICDA-perturbed CpGs and their respective overlap between the three experimental systems are summarized in Supplementary Data 1.

**Epigenetic heterogeneity link to phenotype and transcription.** Given the phenotypic consequence of AICDA overexpression in murine tumors, we investigated whether cytosine methylation was associated with altered gene expression. We performed RNAseq on Vav-*Bcl2*+*Aicda* ($n = 7$) and VavP-*Bcl2* ($n = 6$) tumors and found an expected inverse correlation between gene expression and gene promoter cytosine methylation, especially the region ±250 bp of the transcription start site (TSS) (Supplementary Fig. 7a). The same was noted in *Aicda*[−/−] and WT GCB (Supplementary Fig. 7b) and in AICDA-high and AICDA-low DLBCL cases (Supplementary Fig. 7c). Next, we assessed whether there was a robust gene expression signature distinguishing

VavP-*Bcl2*+*Aicda* tumors compared to their VavP-*Bcl2* controls. We found no evidence of robust gene expression differences between VavP-*Bcl2*+*Aicda* and VavP-*Bcl2* tumors. However, we did find that genes significantly over-representing AICDA signature CpGs in their gene bodies were enriched for lower expression in VavP-*Bcl2*+*Aicda* tumors (gene set enrichment analysis (GSEA) normalized enrichment score (NES) = −2.33, false discovery rate (FDR) < 0.001; Supplementary Fig. 8a). The reciprocal pattern was also observed in GC B cells, where there was higher expression of AICDA gene body signature CpG over-representing genes in *Aicda*[−/−] vs. WT GC B cells (GSEA NES = 1.23, FDR = 0.0397; Supplementary Fig. 8b). DLBCLs are generally classified according to gene expression profiling into GCB DLBCL and activated B-cell-like (ABC) DLBCL[20]. We categorized our AICDA-high and AICDA-low DLBCL cases according to their gene expression profiles (Fig. 4g). Here, we again found that genes significantly over-representing DLBCL AICDA signature CpGs in gene bodies were lower expressed, regardless of whether they were of the ABC subtype (GSEA NES = −1.35, FDR < 0.001; Supplementary Fig. 8c) or GCB subtype (GSEA NES = −1.47, FDR < 0.001; Supplementary Fig. 8d). Pathway analysis of the leading edges from murine and human lymphoma GSEAs (i.e., that exhibit decreased gene expression in AICDA-high tumors) showed enrichment for gene signatures associated with DLBCL, including p53 target genes, MYC repressed genes, and BCL6 target genes, as well as genes involved in GC exit, B-cell receptor signaling, and apoptosis (Supplementary Fig. 8e and Supplementary Data 2). These genes were generally not differentially expressed between ABC-DLBCL and GCB-DLBCL (data not shown). It is not clear however, whether such patterns of AICDA-associated decrease in gene expression contribute to the more aggressive *Aicda*-associated phenotype or whether they simply reflect a dependency of AICDA targeting on RNA polymerase II pausing[21–23].

Despite presumably reflecting late or post-GC B cells, we found that AICDA-high DLBCL were more likely to be classified as ABC subtype than AICDA-low DLBCL (Fisher's exact test $P = 0.021$; Fig. 4h). Using an independent cohort of 287 DLBCL patients, we confirmed that ABC-DLBCL showed higher AICDA expression than GCB-DLBCL (Wilcoxon's $P = 2.6e−08$; Supplementary Fig. 8f). This is consistent with previous reports that AICDA is more highly expressed in ABC-DLBCL[16, 24] and that epigenetic heterogeneity is more pronounced in the ABC subset of DLBCL[3].

Cytosine methylation heterogeneity has been proposed to facilitate chemotherapy resistance[5, 8, 9] Based on this consideration, it might be predicted that particular epigenetic states might be selected after exposure to these drugs. Hence, we investigated whether we could find a reduction in intra-tumor epigenetic heterogeneity in DLBCL at relapse. To this end, we analyzed ERRBS data from a cohort of 13 DLBCL patients, containing paired diagnosis tumors (untreated) and their matched relapsed (chemotherapy-treated) samples[5]. Consistent with the previously reported reduction in epigenetic allele diversity in relapsed cases, our more global analysis of cytosine methylation found that relapsed DLBCL showed a reduction in intra-tumor epigenetic heterogeneity compared to diagnosis samples (Supplementary Fig. 9a). In contrast, diagnosis specimens featured a greater degree of inter-tumor heterogeneity than DLBCLs at diagnosis, as shown both by increased pairwise methylation distance and higher CpG IQR values (Supplementary Fig. 9b, c). Taken together, these different types of analysis suggest that cytosine methylation patterns become more homogenous following chemotherapy treatment, but also that they drift apart from each other. This is consistent with the hypothesis that these tumors have undergone clonal selection (Supplementary Fig. 9d).

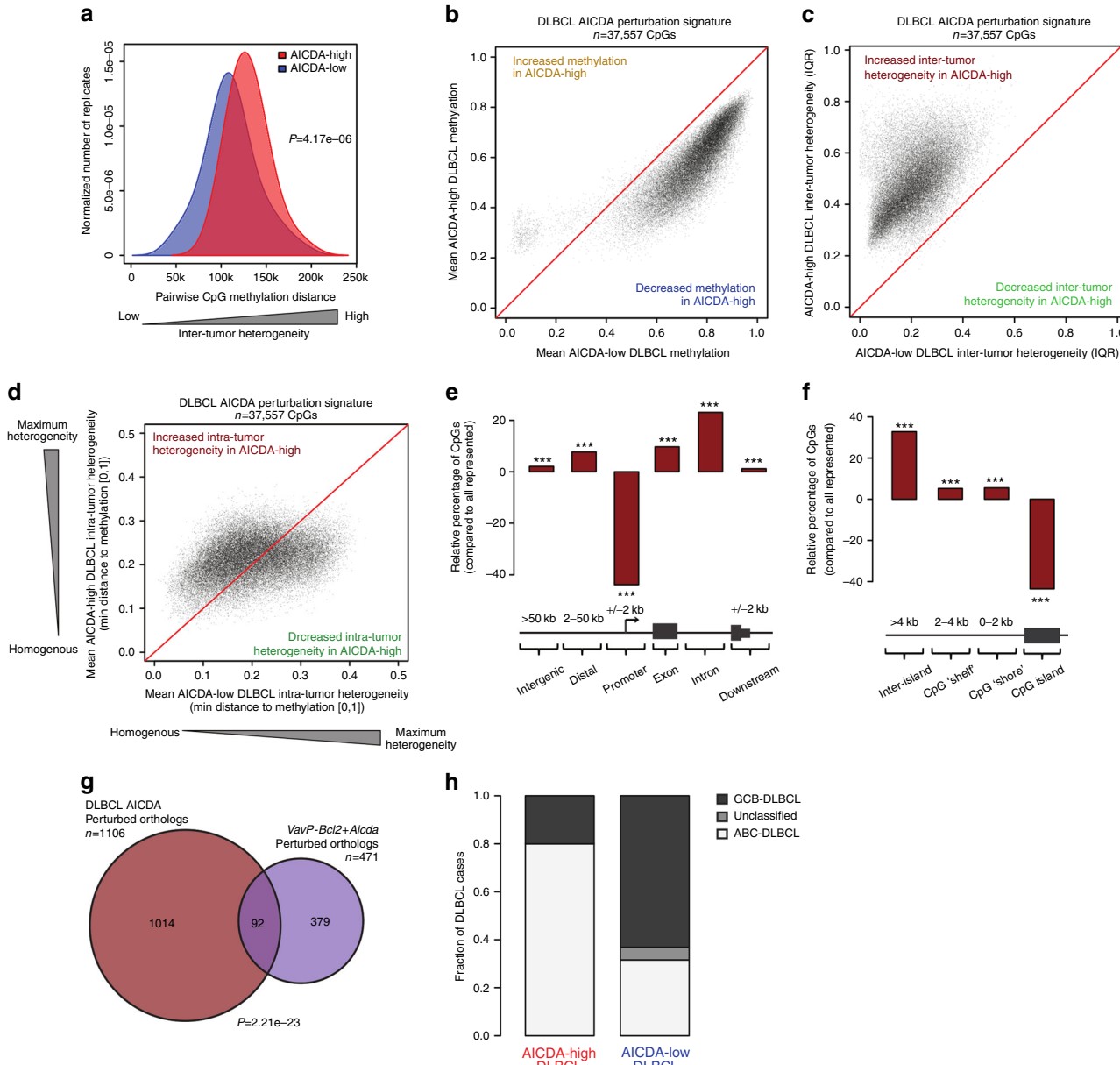

**Fig. 4** High expression of AICDA in DLBCL causes higher DNA methylation heterogeneity and hypomethylation. **a** Density plot showing the inter-tumor pairwise methylation distance between ERRBS profiles of AICDA-low and AICDA-high DLBCL. AICDA-high DLBCL have greater pairwise distance, indicating increased inter-tumor heterogeneity; two-sided Wilcoxon's signed-rank test. **b–d** Scatterplots showing shift in mean methylation (**b**), inter-tumor diversity (**c**), and intra-tumor heterogeneity (**d**) of DLBCL AICDA perturbation signature between AICDA-low and AICDA-high DLBCL. **e** Bar plot showing the distribution of DLBCL AICDA-perturbed CpGs relative to the distribution of all represented CpGs; Fisher's exact test. **f** Bar plot showing the relative distribution of DLBCL AICDA-perturbed CpGs within proximity to CpG islands; Fisher's exact test. **g** Venn diagram showing the overlap between genes significantly over-representing DLBCL AICDA signature CpGs and murine orthologs over-representing VavP-*Bcl2*+*Aicda* signature CpGs; hypergeometric test. **h** Bar plot showing the relative fraction of DLBCL subtypes within AICDA-high and AICDA-low DLBCL cases (*$P < 0.05$, **$P < 0.01$, ***$P < 0.001$)

## Discussion

Besides the role of AICDA in modifying cytosine methylation during the GC reaction[13], the data showed in the present manuscript suggests that AICDA is additionally a critical source of epigenetic heterogeneity in DLBCL. We find that AICDA-linked epigenetic heterogeneity is predominantly associated with relative loss of cytosine methylation, consistent with the known mechanism of action of AICDA in cytosine deamination. Our data suggest a scenario whereby excision of methyl-cytosine is repaired with unmethylated nucleotides. Although AICDA targeting seems to show preference for certain chromatin features, the effect on specific cytosines is likely stochastic, resulting in

gradual divergence of methylation landscapes between cells. Consistent with the known association of cytosine methylation heterogeneity with inferior clinical outcome in human DLBCL patients[3, 4], we found that AICDA overexpression in mice was associated with both increased inter-tumor and intra-tumor methylation heterogeneity (Fig. 2a) and was linked to more aggressive disease in murine B-cell tumors (Fig. 1d). We suggest that AICDA-induced epigenetic heterogeneity increases plasticity, permitting cancer cells a greater degree of population diversity and enhancing the adaptive capacity of the overall tumor. Additionally, we found further evidence that ABC-DLBCLs, which are more aggressive with inferior clinical outcome than

their GCB-DLBCL counterparts[25], are associated with high AICDA expression and greater epigenetic heterogeneity. This suggests that AICDA may be among the contributing factors to epigenetic heterogeneity in ABC-DBCL.

Although the role of AICDA in lymphomagenesis has previously been linked to its mutagenic potential, our data suggest additional dimensions to the deleterious effect of this protein in DLBCL, via enhanced epigenetic plasticity. Consistent with this idea, studies have shown a reduction of cytosine methylation heterogeneity following treatment with chemotherapy, suggestive of clonal selection[5]. The epigenetic role of AICDA in neoplastic transformation and cancer progression may also reach beyond GC-derived lymphomas. AICDA can be transcriptionally upregulated in epithelial cells via nuclear factor-κB signaling following cytokine stimulation or exposure to pathogenic factors, such as *Helicobacter pylori*[26]. AICDA has also been implicated in non-lymphoid cancers, including melanoma and pancreatic carcinomas[27, 28]. Such association between inflammation, infection, and expression of AICDA may prove to be a missing link between chronic inflammation and increased risk of various malignancies.

## Methods

**Animal models**. Bone marrow cells from 8- to 10-week-old BCL2 transgenic animals (VavP-*Bcl2*)[29] were harvested and transduced with viral supernatants containing either EV or AICDA-expressing retroviral vector pMIG, described previously[30, 31]. Two million bone marrow cells of each condition were injected into the tail veins of lethally irradiated recipient C57BL/6J mice. *Aicda*[−/−] mice were a generous gift from T. Honjo. WT (BALB/c) mice were from The Jackson Laboratory. All mice were followed until any one of several criteria for euthanizing were met, including severe lethargy and more than 10% body weight loss in accordance with our Weill Cornell Medicine Institutional Animal Care and Use Committee–approved animal protocols. All animals were maintained according to the guidelines of the Research Animal Resource Center of Weill Cornell Medicine.

**DLBCL patient samples**. Human DLBCL samples were obtained as previously described by Pan et al.[5]. In brief, genomic DNA (gDNA) was extracted from frozen solid tissue sections. The tumor purity of DLBCL samples was above 80–90% based on histological observation. Frozen tissue samples were first digested overnight with 0.5 mg ml$^{-1}$ Proteinase K and 0.625% sodium dodecyl sulfate (SDS) in 4 ml Nucleic Lysis Buffer at 37 °C. After digestion, 1 ml of saturated NaCl was added to the samples and samples were shaken vigorously for 15 s before spun at 2500 r.p.m. for 15 min. Supernatant was transferred to a new tube and mixed with two volumes of 100% ethanol at room temperature. DNA was precipitated by centrifugation at maximum speed for 30 min, washed twice with 70% ethanol, and finally dissolved in TE or nuclease-free water overnight at room temperature.

To assess *AICDA* expression differences between DLBCL subtypes, we analyzed 287 newly diagnosed and characterized DLBCL cases, in which individuals were treated with R-CHOP (given with curative intent) at the BC Cancer Agency (Vancouver)[32]. These studies were approved by the Research Ethics Board at The University of British Columbia, British Columbia Cancer Agency (REB#H13–01478).

**Mouse B-cell isolation**. Spleens of VavP-*Bcl2* and VavP-*Bcl2*+*Aicda* mice were dissected and cell suspensions were prepared. Mononuclear cells were purified using Histopaque (Sigma) gradient centrifugation and B cells were positively selected using anti-B220 magnetic microbeads (Miltenyi Biotech, Germany). For the isolation of GC B cells, WT or *Aicda*[−/−] mice were immunized intraperitoneally with NP-CGG ratio 20–25 (Biosearch Technologies) in alum (1:1) to induce GC formation. Mice were sacrificed at day 10 after immunization, spleens were dissected, and mononuclear cells were purified using Histopaque gradient centrifugation. Cell suspensions were enriched in B cells by positive selection with anti-B220 magnetic microbeads and GC B cells (B220$^+$GL7$^+$FAS$^+$DAPI$^-$) were sorted using a BD FACSAria II sorter.

**Flow cytometry analysis**. Flow cytometry analysis of spleen cell suspensions was performed using the following fluorescent-labeled anti-mouse antibodies: PE-conjugated anti-B220 (BD Pharmingen #553089), PE-Cy7-conjugated anti-CD95 (BD Pharmingen #557653), Alexa Fluor 647-conjugated anti-GL7 (BD Pharmingen #561529), APC-conjugated IgG1 (BD Pharmingen #560089), APC-Cy7-conjugated anti-CD3 (Biolegend #100222), and PE-conjugated anti-Gr1 (eBioscience #12-5931-81). DAPI (4',6-diamidino-2-phenylindole) was used for the exclusion of dead cells. Cell cycle analysis of GC B cells was performed at day 10 after NP-CGG immunization, 2 h after bromodeoxyuridine (BrdU) intraperitoneal injection (2

mg), following the manufacturer's instructions (V450-conjugated anti-BrdU, BD Horizon #560810). Data were acquired on a MACSQuant Flow Cytometer (Miltenyi Biotech) and analyzed using the FlowJo v10.1 software (TreeStar).

**Enhanced reduced representation bisulfite sequencing**. Murine gDNA was extracted using the Puregene Gentra Cell Kit (Qiagen) and eluted in TE. The gDNA quality was assessed using 1% agarose gel to assure no shearing. Quality of purified DNA was checked using an Agilent 2100 Bioanalyzer. Fifty nanograms of 50 ng gDNA was bisulfite converted using the EZ DNA Methylation Kit (Zymo Research). Base-pair resolution DNA methylation analysis was performed on gDNA following the ERRBS protocol previously described[15]. To compare loss of highly methylated CpGs in VavP-*Bcl2*+*Aicda* tumors, we performed a Fisher's exact test and classified "high methylation" state as CpGs with mean methylation level >70%. To quantify the epigenetic heterogeneity within the different replicate conditions, we calculated the pairwise methylation distance between ERRBS profiles using rectilinear Manhattan distance normalized to 1e+06 CpGs. The value of the distribution of all pairwise distances within a group of samples defines the inter-tumor methylation heterogeneity of that group. To make no assumptions regarding the distribution of these pairwise distances, Wilcoxon's rank-sum tests were used to compare between conditions. Delta mean methylation was calculated by subtracting the mean control (e.g., VavP-*Bcl2* tumors, WT GC B cells, AICDA-low DLBCL) methylation value from the mean experimental (e.g., VavP-*Bcl2* +*Aicda* tumors, *Aicda*[−/−] GC B cells, AICDA-high DLBCL) methylation value. Delta IQR among replicates were calculated by subtracting the IQR among control replicates from the IQR among experimental replicates. PC analysis was performed on centered data. AICDA-perturbed CpGs were defined as CpGs with component loading factor >2 standard deviations above mean loading factor. To quantify the degree of intra-sample heterogeneity, we calculated the minimum distance from the methylation value of each CpG to [0,1] (i.e., the distance away from the closest homogenous unmethylated/methylated state). Using this metric, the maximum intra-sample heterogeneity is 0.5, reflecting a population state with half of loci being methylated and the other half being unmethylated. To compare intra-tumor heterogeneity between conditions without assumptions of distribution, we performed pairwise Wilcoxon's rank-sum tests. Genes over-representing signature CpGs were identified according to hypergeometric test (Benjamini–Hochberg-adjusted $P < 0.05$) using all ERRBS-represented CpGs within the gene body (interval from +2 kb of TSS to transcription end site (TES)). Genomic distribution of CpGs was determined using the ChIPseeqerAnnotate module in the ChIPseeqer package[33]. Significance of genomic distributions of AICDA-perturbed CpGs was assessed using a Fisher's exact test vs. all represented, non-perturbed CpGs. To assess a correlation of gene expression with cytosine methylation within promoters, we calculated the mean cytosine methylation within 100 bp bins for interval of −2 kb:+5 kb of TSS. We then categorized all genes according to the expression level: no detectable expression or decile of expression Q1 (lowest) to Q10 (highest), and plotted the mean cytosine methylation for all categorized genes.

**RNA sequencing**. Total RNA was extracted from murine tumors or human DLBCL patient samples using Trizol (Life Technologies) and RNeasy Isolation Kit (Qiagen). RNA concentration was determined using Qubit (Life Technologies) and integrity was verified using Agilent 2100 Bioanalyzer (Agilent Technologies). Libraries were generated using the TruSeq RNA Sample Kit (Illumina). First-strand synthesis was performed using random oligos and SuperscriptIII (Invitrogen). After second-strand synthesis, a 200-bp paired-end library was prepared following the Illumina paired-end library preparation protocol. Pair-end sequencing (PE50) was performed on Illumina HiSeq 2000. RNA sequencing results were aligned to mm10 or hg19, respectively, using STAR[34] and annotated to RefSeq using the Rsubread package[35]. DLBCL subtype classification was performed using RNAseq profiles as described in Cardenas et al.[36]

RNAseq was performed on an independent cohort of 322 fresh frozen-derived RNA samples from the British Columbia Cancer Agency to quantify the gene expression levels of AICDA, as previously described[37]. Between 849 ng and 2 μg of total RNA was used for library preparation. Polyadenylated (polyA+) messenger RNA (mRNA) was purified using the 96-well MultiMACS mRNA Isolation Kit on the MultiMACS 96 separator (Miltenyi Biotec). The libraries were pooled (five libraries per lane) and sequenced as paired-end 75-bp reads on an Illumina HiSeq 2000. Gene expression levels were calculated as fragments per kilobase of exon per million fragments mapped. DLBCL subtype classification for the samples from the British Columbia Cancer Agency was assigned using the Lymph2Cx assay on the nanoString Technologies platform using RNA extracted from formalin-fixed paraffin-embedded biopsies[38, 39].

**Histology and immunohistochemistry**. Mice organs were fixed in 4% formaldehyde and embedded in paraffin. Four micron sections were deparaffinized and heat antigen-retrieved in citrate buffer (pH 6.4) and endogenous peroxidase (HRP) activity was blocked by treating the sections with 3% hydrogen peroxide in methanol. Indirect immunohistochemistry was performed with antispecies-specific biotinylated secondary antibodies followed by avidin–horseradish peroxidase or avidin–alkaline phosphatase, and developed by Vector Blue or DAB color substrates (Vector Laboratories). Sections were counterstained with hematoxylin if

necessary. The following primary antibodies were used: biotin-conjugated anti-B220 (Invitrogen RM2615), CD3 (Vector VP-RM01), and Ki67 (Vector VP-K451). Photomicrographs were scanned using the VENTANA iScan HT scanner and Virtuoso software (Roche). Ki67-positive cells were quantified in 10 high-power microscopic fields at ×60 magnification. A histologic score system consisting of a relative scale of organ infiltration was used, corresponding to none (0), mild (1), moderate (2), and marked (3) infiltration by neoplastic lymphocytes.

**Immunoblotting**. Total cells extracts from splenocytes of VavP-Bcl2 ($n = 6$) and VavP-Bcl2+Aicda ($n = 7$) mice were prepared after treatment with lysis buffer (10 mM Tris, pH 7.4, 150 mM NaCl, 1% NP-40, 0.5% deoxycholate, 1% Triton X-100, 0.1% SDS, and 1 mM EDTA), supplemented with PMSF (Sigma) and a protease inhibitor cocktail (Roche). Lysates were subjected to SDS-PAGE (polyacrylamide gel electrophoresis), transferred to a PVDF (polyvinylidene fluoride) membrane (Bio-Rad Laboratories) and blotted with anti-AICDA (L7E7, Cell Signaling Technology) or anti-actin (A5441, Sigma). Signals were detected with HRP-conjugated secondary antibodies (Santa Cruz Biotechnology) using the ECL system (Thermo Scientific).

**SHM analysis**. We amplified by PCR the genomic areas described previously as AICDA off targets in $J_H4$[40], $S\mu$[41], Bcl6, Cd83, and Pax5[42]. The amplicons were pooled and cleaned-up using the MinElute PCR Purification Kit (Qiagen). Sequencing libraries were constructed from the purified PCR product by using Illumina TruSeq DNA Sample Preparation Kit v2 (Illumina). Each sample was tagged with a unique index and sequenced on the Illumina MiSeq platform producing 2 × 300 bp paired-end reads. PE 2 × 300 MiSeq reads were aligned to the Mus Musculus GRCm28 genome using BWA and pileups were generated using samtools mpileup. The number of mutations, either nonsynonymous (missense, frameshift, premature termination) or indels, were counted, excluding known single-nucleotide polymorphisms (dbSNP release 134) and high variant allele frequency variants (>25%). We quantified mutation burden as the number of mutated bases within the sequenced reads, normalized to the total number of sequenced nucleotides in kilobases.

**Exome sequencing**. Whole exome sequencing was performed on B220$^+$ lymphoma cells from the spleens of VavP-Bcl2 ($n = 4$) and VavP-Bcl2+Aicda ($n = 4$). One microgram of high-molecular-weight genomic DNA was used to prepare exome sequencing libraries using the Aglient SureSelectXT Human All Exon 50 Mb Target Enrichment System for Illumina Pair-End Sequencing Library Kit (Agilent Technologies, Santa Clara, CA, USA). Each library was sequenced on one entire lane of a flow cell on an Illumina HiSeq 2000. Sequence information of 75 bp on each end of the DNA library fragment (PE75) was collected. To analyze mutation burden, exome samples were aligned to the Mus Musculus GRCm38 using STAR 2.4.0[34]. A pileup of sorted bam files was prepared for each gene with quality score >20 using samtools. From each pileup, a list of single-nucleotide polymorphisms was generated at each base with minimum coverage of 10 reads using VarScan 2.3.4[43]. We quantified mutation burden as the number of mutated bases (e.g., nonsynonymous or indels), normalized to the total number of sequenced nucleotides in kilobases, excluding nucleotides that mapped to bases with <10 reads of coverage.

**IgVH rearrangement analysis**. PCR to evaluate IgVH rearrangements was performed on gDNA from B220$^+$ lymphoma cells of VavP-Bcl2 ($n = 3$) and VavP-Bcl2+Aicda ($n = 3$) mice, using primers that annealed to the variable region of the abundantly used gene families $V_HJ558$-$J_H4$ and $V_HGam3.8$-$J_H4$, described previously[44].

**Data availability**. Raw data from RNA sequencing and ERRBS in murine lymphomas and human DLBCL were deposited in GEO under accession number GSE95013. Raw data from RNA sequencing and ERRBS in murine GC B cells are available under accession number GSE71702.

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

## Acknowledgements

We would like to thank T. Honjo (Kyoto University Graduate School of Medicine) for *Aicda*$^{-/-}$ mice, members of Melnick laboratory for useful discussions and suggestions, Epigenomics Core WCM, Genomics Resources Core Facility WCM, Flow Cytometry Core Facility WCM, Laboratory of Comparative Pathology WCMC and MSKCC. P.M.D. is supported by a Lymphoma Research Foundation Post-doctoral Fellowship. O.E. is supported by NSF CAREER, LLS SCOR, Hirschl Trust Award, Starr Cancer Consortium I6-A618, NIH 1R01CA19454. A.M. is supported by the Chemotherapy Foundation, Leukemia and Lymphoma Society SCOR Grant #7012-16 and the Starr Cancer Consortium.

## Author contributions

M.T. and P.M.D. performed the experiments, analyzed the data, and wrote the manuscript; D.R., Z.C., and D. E. contributed to the bioinformatics analysis; D.W.S., P.G., R.D. G., and G.I. provided human primary DLBCL data; J.C. provided reagents, L.C. and I.A. performed whole exome sequencing; O.E., A.M., and R.S. designed the experiments, wrote the manuscript, and supervised the overall study.

## Additional information

**Competing interests:** The authors declare no competing financial interests.

