## [Peer Review File · Nature Communications]

Reviewers' comments:

Reviewer #1 (Remarks to the Author):

In this manuscript, by Teater et al., the authors use an HSC transplantation model and primary human tumors to show an association between high AID expression and epigenetic heterogeneity. The implication is that this heterogeneity may allow for a more diverse pool of tumor cell clones, of which one or more may be resistant to chemotherapy. This is an interesting and comprehensive paper, and I only have minor hesitations at this time.

Comments

1/ The approach used in this paper is to express AID from a retroviral vector in HSC from VavP-Bcl2 mice and use these to reconstitute a recipient mouse. This presumably results in constitutive expression of AID in all haematopoietic compartments. With this being the case, how much of the epigenetic heterogeneity associated with AID expression can be attributed to its activity within haematopoietic progenitors or early stages of B-cell development prior to entering the GC? One would assume that human DLBCL does not express AID at these earlier stages of differentiation and this may therefore confound the comparison of this murine model to the human disease. Was increased epigenetic heterogeneity also seen in non-B-cell lineages originating from these HSC?

2/ Figure 1C shows that the majority of VavP-Bcl2 tumors do not express AID. Were these tumors also of GCB origin? Is there no AID expression, or are the levels relatively lower and not apparent on this western blot?

3/ The observation that reduced gene body methylation was associated with reduced gene expression is an interesting one. But what were the functions of those genes that showed decreased gene body methylation and reduced expression in tumors? Are these genes that have been implicated in lymphoma? Are they differentially expressed in GCB vs ABC?

4/ Although the hypothesis makes sense, there is no experimental evidence that increased epigenetic heterogeneity drives chemotherapy resistance. If these tumors are treated with chemotherapy, is there a reduction in the epigenetic heterogeneity that would be suggestive of clonal selection?

Reviewer #2 (Remarks to the Author):

In this paper Teater and colleagues test for a functional impact of AICDA overexpression on tumour DNA methylation pattern and disease outcome in BCL2-driven murine lymphomas. They report that AICDA overexpression phenotype is associated with increased cytosine methylation heterogeneity which, in turn, is accompanied by poor disease prognosis. Interestingly, the authors report that AICDA driven hypomethylation was not accompanied by an increased burden of somatic mutations.

Although this study is largely descriptive, I believe establishing AICDA as an important contributor in shaping normal GC B cell and GC tumour epigenetic landscape is of much interest. I enjoyed reading this paper. There are however a number of minor issues that, if properly addressed, would strengthen the manuscript considerably.

1. I was struggling to locate the raw RNA-Seq data from a cohort of 63 primary human DLBCL samples that authors describe on page 7. These data would either need to be uploaded to GEO (as required by NPG), or their alternative location needs to be defined.

2. Is there a functional correlation between DNA methylation and gene expression profiling in authors' system? Since both ERRBS and RNA-Seq datasets are there, the authors should be in position to answer this question.

3. Figure 3f. The Venn diagram overlap between VavP-Bcl2 and GCB AICDA perturbed genes is

rather modest. Can authors comment on the identity of these 172 genes that they see overlapping? Do they represent at least some of the known non-immune targets of AID? I believe this additional bioinformatics work will substantially increase the level of evidence of this manuscript and provide further insight into functional components of the system.

Jun 15, 2017

Manuscript Number: NCOMMS-17-06028-T

"AICDA drives epigenetic heterogeneity and accelerates germinal center-derived lymphomagenesis"

Reviewers' comments:

Reviewer #1 (Remarks to the Author):

In this manuscript, by Teater et al., the authors use an HSC transplantation model and primary human tumors to show an association between high AID expression and epigenetic heterogeneity. The implication is that this heterogeneity may allow for a more diverse pool of tumor cell clones, of which one or more may be resistant to chemotherapy. This is an interesting and comprehensive paper, and I only have minor hesitations at this time.

Comments

1/ The approach used in this paper is to express AID from a retroviral vector in HSC from VavP-Bcl2 mice and use these to reconstitute a recipient mouse. This presumably results in constitutive expression of AID in all haematopoietic compartments. With this being the case, how much of the epigenetic heterogeneity associated with AID expression can be attributed to its activity within haematopoietic progenitors or early stages of B-cell development prior to entering the GC? One would assume that human DLBCL does not express AID at these earlier stages of differentiation and this may therefore confound the comparison of this murine model to the human disease. Was increased epigenetic heterogeneity also seen in non-B-cell lineages originating from these HSC?

Author Response: We are grateful to the reviewer for the constructive comments regarding our study. Following the reviewer's suggestion, we investigated whether AICDA overexpression in HSC was associated with increased global epigenetic heterogeneity in the cells derived from these hematopoietic precursors. To test this, we analyzed ERRBS profiles from two non-B cell populations that were differentiated from the transplanted bone marrow cells in the VavP-Bcl2 cohort. We isolated T cells and myeloid cells (Gr1⁺ cells) from the spleens of transplanted mice, which received either control (VavP-Bcl2, n=3) or AICDA-overexpressing (VavP-Bcl2+Aicda, n=3) BCL2 transgenic cells (Rebuttal Figure 1a, top). We also sorted hematopoietic progenitors (Lin⁻Sca-1⁺c-Kit⁺(LSK) cells) from the bone marrow of these mice, although unfortunately the amount of DNA that we obtained was insufficient to perform ERRBS (i.e. below 35ng, **Rebuttal Figure 1a**, bottom).

We first compared the global ERRBS profiles of B cell lymphomas, Gr1⁺ cells, and T cells. We found that, whereas VavP-Bcl2+Aicda B cell tumors showed a reduction in the proportion of CpGs with high methylation (e.g. >80%), neither the VavP-Bcl2+Aicda Gr1⁺ cells nor the VavP-Bcl2+Aicda T cells showed dramatic reduction (**Rebuttal Fig. 1b**). Indeed the VavP-Bcl2+Aicda Gr1⁺ cells even showed some increase in the highly methylated fraction of CpGs.

Next, to even more directly address the reviewer question, we analyzed inter-individual methylation heterogeneity using pairwise methylation distance, similar to what was performed in the various B-cell experiments reported in this paper. While AICDA-overexpressing B-cell tumors exhibited increased inter-tumor heterogeneity compared to control tumors, as shown in the manuscript (Figure 2a), both

VavP-*Bcl2*+*Aicda* Gr1+ cells and T cells showed no significant difference in inter-individual methylation heterogeneity compared to control cells (Rebuttal Fig. 1c; Gr1+ Wilcoxon p=0.0571, T cell Wilcoxon p=0.070).

In conclusion, we find that AICDA overexpression does not induce the same degree of hypomethylation or methylation heterogeneity in differentiated myeloid or T cells. The increased cytosine methylation heterogeneity observed in AICDA-overexpressing B-cell tumors is specific to GC-derived cells and is not due to inheritance of methylation patterns associated with early stages of hematopoietic development.

Rebuttal Figure 1. Non-B cell types in VavP-Bcl2 mice do not show reduction of methylated cytosines or increase in methylation heterogeneity. (a) Flow cytometry gating strategy to isolate T cells and Gr1+ cells from the spleen (top) and LSK from bone marrow (bottom) of VavP-Bcl2 and VavP-Bcl2+Aicda mice. The number of sorted cells and the amount of gDNA obtained are also shown. (b) Density plots showing the mean methylation values in murine tumors (left), Gr1+ cells (middle), and T cells (right). Inset magnifies the density at highly methylated CpGs (>80%). (c) Density plot showing the pairwise methylation distance for Gr1+ cells (left) and T cells (right).

Action taken: We have mentioned these new results in the text (Page 5) as data not shown (as these are basically negative results), but would be happy to include this figure if the editor and reviewer think it is necessary.

2/ Figure 1C shows that the majority of VavP-Bcl2 tumors do not express AID. Were these tumors also of GCB origin? Is there no AID expression, or are the levels relatively lower and not apparent on this western blot?

Author Response: We thank the reviewer for this comment. To address this question, we repeated the western blot loading a higher amount of protein (**Rebuttal Figure 2a**). It can now be better appreciated that all the samples express AICDA, consistent with the GC origin of these tumors, although the protein level is generally lower in VavP-Bcl2 tumors than in VavP-Bcl2+Aicda tumors. This result is supported by our flow cytometry data, since the proportion of GC B cells (B220⁺ CD95⁺ GL7^{int/high}) in all the tumor samples (VavP-Bcl2 and VavP-Bcl2+Aicda) was significantly higher than in WT immunized mice with sheep red blood cells (SRBC, **Rebuttal Figure 2b** and **Supplementary Fig. 1d**). As expected given the increased tumor burden, the abundance of GC-derived tumor cells was significantly higher in the VavP-Bcl2+Aicda mice.

Rebuttal Figure 2. VavP-Bcl2 tumors express AICDA and germinal center immunophenotypic markers. (a) Immunoblot of AICDA and b-actin in splenocytes from VavP-Bcl2 (n=6) and VavP-Bcl2+Aicda (n=7) mice, loading 45mg (old) or 100mg (new) of protein. (b) Representative flow cytometry plot (left) and quantification (right) showing the percentage of B220+CD95+ cells in the spleen of VavP-Bcl2 and VavP-Bcl2+Aicda mice compared to WT mice immunized with sheep red blood cells for 10 days.

Action taken: We have modified Supplementary Figure 1 to include these results and we have clarified this point in the corresponding section of the results (Page 4).

3/ The observation that reduced gene body methylation was associated with reduced gene expression is an interesting one. But what were the functions of those genes that showed decreased gene body methylation and reduced expression in tumors? Are these genes that have been implicated in lymphoma? Are they differentially expressed in GCB vs ABC?

Author Response: In response to this important question, we have performed a pathway analysis using the GSEA leading edge genes from VavP-Bcl2+Aicda (**Supplementary Figure 6a**) as well as the genes included in GSEA leading edge for both human DLBCL subtypes (**Supplementary Figure 6c,d**). We find that these genes are enriched for several gene signatures related to DLBCL including p53 target genes, MYC repressed genes, BCL6 target genes, as well as genes involved in GC exit, BCR signaling, and apoptosis (**Rebuttal Figure 3a**). Examining the list of genes involved revealed many genes encoding tumor suppressors potentially linked to lymphomagenesis such as *IRF4*, *EBF1*, and *PTPN13*¹⁻³. The full list of genes is included in **new Supplementary Table 2**. Using edgeR and an independent cohort of DLBCL patients subtyped according to gene expression profiling, we assessed whether the leading edge genes from human DLBCL were differentially expressed in ABC-DLBCL vs GCB-DLBCL. We found that the majority of the leading edge genes were not differentially expressed

(195 of 234), but that 5 genes were downregulated (hypergeometric $P=0.0732$) and that 34 genes were up-regulated (hypergeometric $P=6.58e-04$; **Rebuttal Fig. 3b**).

Rebuttal Figure 3. AICDA-perturbed, downregulated genes in lymphoma. (a) Heatmap showing enrichment significance of GSEA leading edge genes for gene signatures identified by various databases; the color key shows statistical significance based on hypergeometric test. (b) Volcano plot showing fold-change and significance of genes between ABC- and GCB-DLBCL profiles. Red dots indicate leading edge from DLBCL AICDA-perturbed gene GSEAs.

Action taken: We have included the new pathway analysis into Supplementary Figure 7 (Supplementary Fig. 7e), created a new Supplementary Table 2 and added the related text to page 9. The negative result about the differential expression between GCB- and ABC-subtypes is mentioned on page 9 as data not shown.

4/ Although the hypothesis makes sense, there is no experimental evidence that increased epigenetic heterogeneity drives chemotherapy resistance. If these tumors are treated with chemotherapy, is there a reduction in the epigenetic heterogeneity that would be suggestive of clonal selection?

Author Response (see rebuttal Figures 4 and 5): We are grateful to the reviewer for this excellent question. Previous work from our group⁴ profiling thirteen DLBCL diagnosis-relapse pairs showed decreased epipolymorphisms (a measure of epigenetic allele diversity) in relapse versus diagnosis (**Rebuttal Figure 4a**). Additionally, they showed that DLBCL patients with higher versus lower intra-tumor heterogeneity had worse progression-free survival (**Rebuttal Figure 4b**).

Rebuttal Figure 4. Intra-tumor heterogeneity in DLBCL diagnosis-relapse pairs. (a) Median epipolymorphism for diagnosis and relapse tumors from representative patient in DLBCL cohort. Intra-tumor methylation heterogeneity is visibly reduced after therapy. (b) Kaplan-Meier plot comparing the progression-free survival between DLBCL patients with higher (30%, n=18) versus lower (30%, n=18) intra-tumor methylation heterogeneity in an independent DLBCL cohort. Figure adapted from Pan, et al. Nat Comm 2015.

However it is important to underline that the approach from Pan et.al. is different from that used for this manuscript. The Pan et.al. analysis was based on cytosine methylation heterogeneity as measured by epipolymorphism level of sets of 4 consecutive CpGs (i.e. epigenetic alleles). Given the global depletion and clustering of CpG dinucleotides in the human genome, this analysis will overwhelmingly interrogate genomic regions near promoters. In contrast, our more comprehensive approach was agnostic to CpG localization since, unlike the epipolymorphism method, we are not restricted to only analyzing Illumina reads containing 4 consecutive CpGs.

Therefore, to further address the reviewer's questions and better assess cytosine methylation heterogeneity genome-wide, we used the more global genomic approach employed in our current manuscript to compare and contrast intra-tumor heterogeneity (minimum methylation difference from [0,1]) in these paired diagnosis and relapse DLBCL cases. Consistent with the observations of Pan et al., we found that relapsed DLBCL showed a reduction in intra-tumor heterogeneity compared to diagnosis DLBCL (**Rebuttal Figure 5a**).

It is important to note that the Pan et.al. paper did not at all explore inter-tumor heterogeneity as we have done in this manuscript. We therefore also examined this aspect of the data (pairwise methylation distance and distribution of IQR for all common CpGs) and found in contrast that relapsed DLBCL exhibited greater inter-tumor heterogeneity, as shown both by increased pairwise methylation distance and higher CpG IQR values vs the diagnostic specimens (**Rebuttal Figure 5b,c**).

Taken together, these different types of analysis suggest that cytosine methylation patterns become more homogenous following relapse, but also that they drift apart from each other, which is an intriguing new observation that we would have missed without the reviewer's excellent suggestion. This is consistent with the hypothesis that these tumors have undergone clonal selection (**Rebuttal Figure 5d**). These interesting new results have been added to the paper as Supplementary Figure 8 and are discussed in the text (Page 9).

Rebuttal Figure 5. Relapsed DLBCL have decreased intra-tumor heterogeneity and increased inter-tumor heterogeneity compared to diagnosis. (a) Density plot showing mean intra-tumor heterogeneity of global diagnosis and relapse DLBCL ERRBS profiles. Intra-tumor heterogeneity density is shifted to the left, indicating a reduction compared to diagnosis. (b-c) Density plots showing pairwise methylation distance of DLBCL profiles (b) and IQR of all commonly represented CpGs among respective diagnosis and relapsed profiles (c). In both cases, the density is shifted to the right, indicating an increase in inter-tumor heterogeneity. (d) Illustration depicting how lymphoma cells are more heterogeneous than normal cells and exhibit both intra and inter-tumor heterogeneity, whereas at relapse they manifest lower intra-tumor heterogeneity but have drifted further apart (increased inter-tumor heterogeneity).

Action taken: We have generated a new Supplementary Figure 8 and added the information to the text in page 9 as noted above.

Reviewer #2 (Remarks to the Author):

In this paper Teater and colleagues test for a functional impact of AICDA overexpression on tumour DNA methylation pattern and disease outcome in BCL2-driven murine lymphomas. They report that AICDA overexpression phenotype is associated with increased cytosine methylation heterogeneity which, in turn, is accompanied by poor disease prognosis. Interestingly, the authors report that AICDA driven hypomethylation was not accompanied by an increased burden of somatic mutations. Although this study is largely descriptive, I believe establishing AICDA as an important contributor in shaping normal GC B cell and GC tumour epigenetic landscape is of much interest. I enjoyed reading this paper. There are however a number of minor issues that, if properly addressed, would strengthen the manuscript considerably.

1. I was struggling to locate the raw RNA-Seq data from a cohort of 63 primary human DLBCL samples that authors describe on page 7. These data would either need to be uploaded to GEO (as required by NPG), or their alternative location needs to be defined.

Author Response: We are grateful to the reviewer for noticing this missing information.

Action taken: DLBCL RNAseq and ERRBS data for AICDA-high and AICDA-low DLBCL cases were submitted to GEO and are available under the accession number GSE95013.

2. Is there a functional correlation between DNA methylation and gene expression profiling in authors' system? Since both ERRBS and RNA-Seq datasets are there, the authors should be in position to answer this question.

Author Response: We thank the reviewer for raising this question. To assess the functional correlation between cytosine methylation and gene expression, we evaluated the mean cytosine methylation value of genes within 100bp bins from 2kb upstream to 5kb downstream of TSS. By comparing the cytosine methylation pattern according to gene expression: no expression ("FPKM_0") or expression deciles (Q1 – Q10, from lowest to highest), we found (as expected) a clear inverse correlation between gene expression and cytosine methylation in gene promoters, especially in the region +/-250bp of TSS, as has been observed consistently in the literature. This association was uniformly detected across all three systems, both in control and AICDA-modulated samples (**Rebuttal Figure 6a, b, c**). Outside of promoter regions, there was no clear correlation between DNA methylation and gene expression.

Rebuttal Figure 6. Cytosine methylation at TSS is associated with low gene expression, but not with AICDA-associated gene expression changes. (a) Plots showing mean cytosine methylation of gene promoters according to their expression in VavP-Bcl2 tumors (top) and VavP-Bcl2+Aicda tumors (bottom); “FPKM_0” indicates no expression, Q1-Q10 indicates deciles of expression from lowest to highest. (b) Plot showing mean cytosine methylation of gene promoters according to their expression in WT GC B cells (top) and *Aicda*^{-/-} GC B cells (bottom). (c) Plot showing mean cytosine methylation of gene promoters according to their expression in AICDA-low (top) and AICDA-high DLBCL (bottom).

Action taken: We have added this figure as Supplementary Figure 6 in the manuscript and added the information to the text on page 8.

3. Figure 3f. The Venn diagram overlap between VavP-Bcl2 and GCB AICDA perturbed genes is rather modest. Can authors comment on the identity of these 172 genes that they see overlapping? Do they represent at least some of the known non-immune targets of AID?

Author Response. We thank the reviewer for this comment. To address this point we performed a pathway analysis using the lists of AICDA-perturbed from the Venn diagram in **Figure 3f**. We find that these genes are enriched for several gene signatures relevant to GCs and lymphoma, including signatures associated with Myc, p53, bcl6, hypoxia, and GC exit (**Rebuttal Figure 7a**).

To address the second part of the question we identified AICDA-perturbed genes that have previously been reported either as somatic hypermutation hotspots⁵, recurrent sites of aberrant somatic hypermutation in lymphoma⁶, or AICDA off-target sites identified using high-throughput, genome-wide translocation sequencing⁷. Of these 178 genes that have been implicated as a non-Ig target of AICDA, we found that 14 were found to over-represent AICDA-perturbed CpGs in either the GC or VavP-Bcl2 systems. We have incorporated these 14 genes into a Venn diagram to illustrate where they occur (**Rebuttal Figure 7b**).

Rebuttal Figure 7. Role of AICDA-perturbed genes. (a) Heatmap showing enrichment significance of AICDA-perturbed genes for signatures identified by various databases; hypergeometric test. (b) Venn diagrams showing the overlap between genes significantly over-representing VavP-Bcl2+Aicda signature CpGs and GC Aicda-/- signature CpGs. Previously described AICDA non-Ig off-targets are indicated.

Action taken: We have added the pathway heatmap to Supplementary Figure 4 (**Supplementary Fig. 4d**). The genes included in this pathway analysis are listed in **Supplementary Table 1**. The overlap with known AICDA somatic hypermutation off-targets is mentioned in the manuscript text (Page 7).

I believe this additional bioinformatics work will substantially increase the level of evidence of this manuscript and provide further insight into functional components of the system.

List of References

1. Morin, R. D. *et al.* Frequent mutation of histone-modifying genes in non-Hodgkin lymphoma. *Nature* **476**, 298–303 (2011).
2. Mullighan, C. G. *et al.* Genome-wide analysis of genetic alterations in acute lymphoblastic leukaemia. *Nature* **446**, 758–764 (2007).
3. Zhao, S., Sedwick, D. & Wang, Z. Genetic alterations of protein tyrosine phosphatases in human cancers. *Oncogene* **34**, 3885–3894 (2014).
4. Pan, H. *et al.* Epigenomic evolution in diffuse large B-cell lymphomas. *Nature Communications* **6**, 1–12 (2015).
5. Liu, M. *et al.* Two levels of protection for the B cell genome during somatic hypermutation. *Nature* **451**, 841–845 (2008).
6. Khodabakhshi, A. H. *et al.* Recurrent targets of aberrant somatic hypermutation in lymphoma. *Oncotarget* **3**, 1308–1319 (2012).
7. Meng, F.-L. *et al.* Convergent transcription at intragenic super-enhancers targets AID-initiated genomic instability. *Cell* **159**, 1538–1548 (2014).

REVIEWERS' COMMENTS:

Reviewer #1 (Remarks to the Author):

Teater et al., have performed multiple experiments to address my comments from the first round of review. This extensive work has addressed all of my original concerns and significantly strengthened the manuscript.

Minor additional comments:

1/ It would be beneficial to include Rebuttal Figure 1 as a supplementary data item. This was my main concern over this model, and I believe that debunking this concern in a supplementary item is an important step to providing validity to the remaining experiments – particularly if the authors opt out of publishing the peer review material with their manuscript.

2/ There is a typo on line 6 of page 6 (off vs of).

Reviewer #2 (Remarks to the Author):

The authors have satisfactorily addressed all the reviewers' comments and I have no reservations about publishing this manuscript in Nature Communications. Good Job.

Oct 15, 2017

Manuscript Number: NCOMMS-17-06028-T

“AICDA drives epigenetic heterogeneity and accelerates germinal center-derived lymphomagenesis”

Reviewers' comments:

Reviewer #1 (Remarks to the Author):

Teater et al., have performed multiple experiments to address my comments from the first round of review. This extensive work has addressed all of my original concerns and significantly strengthened the manuscript.

Minor additional comments:

1/ It would be beneficial to include Rebuttal Figure 1 as a supplementary data item. This was my main concern over this model, and I believe that debunking this concern in a supplementary item is an important step to providing validity to their remaining experiments – particularly if the authors opt out of publishing the peer review material with their manuscript.

Author Response: We thank the reviewer for this comment. We have added these results as Supplementary Figure 3 and have revised the text accordingly.

2/ There is a typo on line 6 of page 6 (off vs of).

Author Response: We are grateful to the author for noticing this error. We have corrected the text.

Reviewer #2 (Remarks to the Author):

The authors have satisfactorily addressed all the reviews' comments and I have no reservations about publishing this manuscript in Nature Communications. Good job.

Author Response: We thank the reviewer for their cooperation and kind words.